# Novel Therapeutics for Type 2 Diabetes Mellitus—A Look at the Past Decade and a Glimpse into the Future

**DOI:** 10.3390/biomedicines12071386

**Published:** 2024-06-21

**Authors:** Ying Jie Chee, Rinkoo Dalan

**Affiliations:** 1Department of Endocrinology, Tan Tock Seng Hospital, Singapore 308433, Singapore; rinkoo_dalan@ttsh.com.sg; 2Lee Kong Chian School of Medicine, Nanyang Technological University, Singapore 308232, Singapore

**Keywords:** type 2 diabetes mellitus, cardiovascular, renal, sodium-glucose co-transporter-2 inhibitors, glucagon-like peptide-1 receptor agonists, novel agents

## Abstract

Cardiovascular disease (CVD) and kidney disease are the main causes of morbidity and mortality in type 2 diabetes mellitus (T2DM). Globally, the incidence of T2DM continues to rise. A substantial increase in the burden of CVD and renal disease, alongside the socioeconomic implications, would be anticipated. Adopting a purely glucose-centric approach focusing only on glycemic targets is no longer adequate to mitigate the cardiovascular risks in T2DM. In the past decade, significant advancement has been achieved in expanding the pharmaceutical options for T2DM, with novel agents such as the sodium-glucose cotransporter type 2 (SGLT2) inhibitors and glucagon-like peptide receptor agonists (GLP-1 RAs) demonstrating robust evidence in cardiorenal protection. Combinatorial approaches comprising multiple pharmacotherapies combined in a single agent are an emerging and promising way to not only enhance patient adherence and improve glycemic control but also to achieve the potential synergistic effects for greater cardiorenal protection. In this review, we provide an update on the novel antidiabetic agents in the past decade, with an appraisal of the mechanisms contributing to cardiorenal protection. Additionally, we offer a glimpse into the landscape of T2DM management in the near future by providing a comprehensive summary of upcoming agents in early-phase trials.

## 1. Introduction

Individuals with type 2 diabetes mellitus (T2DM) have at two- to four-fold [1] increased risk of cardiovascular disease (CVD). Globally, nearly one-third of people with T2DM are affected by CVD [2]. CVD accounts for approximately half of all mortalities globally [3], disproportionately affecting individuals with T2DM to a much greater extent than individuals without T2DM. According to the United Kingdom Prospective Diabetes (UKPDS) Study follow-up data, fatal CVD is 70 times more frequent than microvascular complications after nine years of follow-up for T2DM [4].

Good glycemic control remains one of the fundamental cornerstones in managing T2DM, alongside cardiovascular risk reduction. Although intensive glycemic control is known to reduce microvascular and cardiovascular complications in type 1 DM [1,5], achieving satisfactory glycemic control alone is inadequate in lowering the cardiovascular burden in T2DM. As such, the US Food and Drug Administration and European Medicines Agency mandate that all new antidiabetic drugs be rigorously evaluated for cardiovascular safety and benefits through cardiovascular outcome trials [6]. 

The past decade witnessed the completion of the landmark cardiovascular outcome for sodium-glucose co-transporter-2 (SGLT2) inhibitors and glucagon-like peptide-1 (GLP-1) receptor agonists (GLP-1 RA). The SGLT2 inhibitor empagliflozin and the GLP-1 RA liraglutide were evaluated in the EMPA-REG [6] and LEADER [7] cardiovascular outcome trials (CVOTs), respectively. The significant cardiovascular risk reduction observed in these landmark trials established a new paradigm in T2DM management, shifting the traditional glucose-centric approach to one that emphasizes cardiovascular risk reduction. Although designed as antidiabetic agents, the cardiovascular and renal benefits conferred by SGLT2 inhibitors and GLP-1 RAs extended the indications to individuals with heart failure and kidney disease without diabetes [8,9,10]. Nearly a decade has passed since the release of the landmark EMPA-REG trial. In the past decade, the global use of SGLT2 inhibitors and GLP-1 RAs has increased. The DISCOVER study, which assessed global trends in SGLT2 inhibitor and GLP-1 RA prescriptions in more than 14,000 patients with T2DM from 37 countries, reported an increase in SGLT2 inhibitor and GLP-1 RA use from 10.8% to 16.1% over 8 years [11]. With the emergence of more CVOT data, the usage of SGLT2 inhibitors and GLP-1 RAs is expected to increase. 

In the next decade, breaking new frontiers in discovering novel pathways and pharmaceutical targets for glucose-lowering pathways will be an important focus in reducing the global burden of T2DM. Given the vascular-centric nature of T2DM, with the process of atherosclerotic damage starting even before the development of diabetes [12], it will be crucial to understand the impact of these medications on the early atherosclerotic processes and progression over time. Novel antidiabetic agents will need to be rigorously evaluated through CVOTs. We scope the literature to review the evidence by which these medications can reverse the process of vascular dysfunction, and subsequent cardiorenal outcomes. This review aims to provide a comprehensive discussion of the past, present and future of T2DM management. We focus on the evidence proving cardiovascular and renal protection is conferred by the newer antidiabetic agents. We appraise the current landscape of novel antidiabetic agents that are currently awaiting further evaluation through well-designed CVOTs and offer insights into pharmaceutical targets for T2DM that are in early-phase clinical trials. 

## 2. Materials and Methods

Our first goal is to appraise the literature for human and animal model studies that evaluate the efficacy and provide mechanistic insights into the cardiovascular and renal benefits of the relatively novel classes of antidiabetic medications available in the past decade. Our second goal is to provide an updated appraisal of candidate agents in early-phase trials with potential cardiovascular benefits. We searched MEDLINE using the following search terms: 

Search terms 1—to identify novel agents:

(“diabetes mellitus, type 2”[MeSH Terms] OR type 2 diabetes[Text Word] OR “ïnsulin resistance”) AND (“novel agent*” OR “emerging treatment*” OR “emerging therapy*” OR “new agent” OR “new treatment*” OR “new therap*”) to screen for novel agents in type 2 DM. 

Search terms 2—to identify relevant human and animal studies on SGLT2 inhibitors and GLP-1 RAs:

For SGLT2 inhibitor: (“SGLT2 inhibitor” OR empaglifozin OR dapagliflozin OR canagliflozin OR ertugliflozin OR ipragliflozin OR sotagliflozin OR tofogliflozin) AND (“endothelial function” OR “flow-mediated dilation” OR “arterial stiffness” OR “pulse wave velocity” OR “cardio-ankle vascular index” OR “CAVI” OR “CRP” or “hsCRP” OR “C-reactive protein” OR “endothelial adhesion molecule” OR “thrombosis marker*” OR “thrombotic marker*” OR biomarker* OR interleukin) 

For GLP-1 RAs: (“glucagon like peptide-1 receptor agonist” OR liraglutide OR semaglutide OR dulaglutide OR albiglutide OR exenatide OR lixisenatide OR tirzepatide) AND (“endothelial function” OR “flow mediated dilation” OR “arterial stiffness” OR “pulse wave velocity” OR “cardio-ankle vascular index” OR “CAVI” OR “CRP” or “hsCRP” OR “C-reactive protein” OR “endothelial adhesion molecule” OR “thrombosis marker*” OR “thrombotic marker*” OR biomarker* OR interleukin OR nephropathy OR renal OR kidney). 

The timeline for all searches is restricted to 10 years. 

We also searched for early-phase trials on novel antidiabetic agents on ClinicalTrials.gov. We only use the search term type 2 diabetes mellitus and apply additional filters to identify studies whose status is planned but not yet recruiting, recruiting, enrolling by invitation, and active, or not recruiting. We limited the age group to adults 18 years and above. 

Our searches yielded the following:

Search terms 1 yielded 1936 results. 

Search terms 2 (SGLT2 inhibitors) yielded 2788 results. 

Search terms 3 (GLP-1 Ras) yielded 1381 results. 

ClinialTrials.gov yielded 414 trials.

As this narrative review aims to comprehensively search the literature for updates in novel pharmacological therapies in T2DM and identify candidate agents in early-phase trials, we did not set the inclusion and exclusion criteria to limit the search. We summarize existing literature to provide an updated overview of the landscape of novel pharmacological therapies in T2DM with cardiovascular and renal benefits. We also discuss the mechanisms and candidate agents in early-phase trials. 

## 3. SGLT2 Inhibitors

In 2015, the release of the EMPA-REG outcome trial demonstrated for the first time that a glucose-lowering agent, empagliflozin, an SGLT2 inhibitor, reduced cardiovascular mortality and heart failure hospitalization when given on top of standard care to patients with T2DM with CVD [6]. Since then, other SGLT2 inhibitors emerged in the market, including canagliflozin [13] and dapagliflozin [14], demonstrating similar class effects in terms of cardioprotection. The cardiovascular event curves begin to separate approximately 3–6 months after initiation of an SGLT2 inhibitor. Given the rapidity of cardiovascular risk reduction, it is postulated that there are other mechanisms beyond and independent of improvement in glycemic control. 

The main SGLT2 inhibitors evaluated in cardiovascular outcome trials include empagliflozin (EMPA-REG) [6], canagliflozin (CREDENCE) [13], dapagliflozin (DECLARE-TIMI 58) [14] and ertugliflozin (VERTIS-CV) [15]. The EMPA-REG trial, which recruited individuals with T2DM and established CVD, reported a 14% relative risk reduction (hazard ratio (HR), 0.86; 95% confidence interval (CI), 0.74–0.99) in three-point major adverse cardiovascular events (MACE) comprising of cardiovascular mortality, nonfatal myocardial infarction or nonfatal stroke [6]. The therapeutic benefit of SGLT2 inhibitors in heart failure was first signaled in the EMPA-REG trial, which reported a significant reduction in heart failure hospitalizations by 35% (HR 0.65; 95% CI, 0.5–0.85) in the empagliflozin arm as a secondary outcome [6]. The CREDENCE [13] and DECLARE-TIMI 58 [14] trials validated this observation. Dedicated outcome trials focusing on heart failure patients were conducted to establish the efficacy of SGLT2 inhibitors in managing heart failure. The DAPA-HF trial, which enrolled 4744 patients with stable heart failure with a reduced ejection fraction (HFrEF), reported a reduction in the primary composite outcome of cardiovascular mortality and heart failure hospitalizations by 26% (HR 0.74; 95% CI, 0.65–0.85) [16]. The EMPEROR REDUCED trial, which evaluated empagliflozin, showed a 25% risk reduction (HR 0.75; 95% CI, 0.65–0.86) in the primary composite outcome of cardiovascular mortality and heart failure hospitalizations compared to placebo in individuals with HFrEF [17]. In heart failure with a preserved ejection fraction (HFpEF), SGLT2 inhibitors have shown similar benefits. The EMPEROR-Preserved trial, which investigated the efficacy of empagliflozin in patients with HFpEF and mildly reduced EF irrespective of diabetes, showed a 21% relative risk reduction in the primary composite outcome of cardiovascular mortality and heart failure hospitalizations (HR 0.79; 95% CI, 0.69–0.90) [18]. A similar result was reported in the DELIVER trial, which evaluated dapagliflozin and observed an 18% reduction in the primary composite outcome or worsening heart failure or cardiovascular death (HR 0.82; 95% CI, 0.73–0.92) and heart failure hospitalizations (HR 0.79; 95% CI, 0.69–0.91) [19]. The VERTIS-CV trial, which recruited individuals with T2DM and established CVD, showed that ertugliflozin was non-inferior to placebo in reducing MACE [15]. Figure 1 summarizes key cardiovascular outcome trials (CVOTs) with SGLT2 inhibitors.

Several mechanisms could explain the cardiovascular benefits of SGLT2 inhibitors. These include improvement in vascular dysfunction [20], inflammation [21], oxidative stress [22], and thrombosis [23]. In terms of cardiac markers, SGLT2 inhibitors reduce troponin I [24] and NT-proBNP [25,26]. The cardiovascular protection conferred by SGLT2 inhibitors is summarized in Figure 2 below. 

### 3.1. Effects of SGLT2 Inhibitors on Vascular Function

Subclinical vascular dysfunction, which includes endothelial dysfunction [27] and arterial stiffness [28], is an early process in atherosclerosis. The DEFENCE study, conducted in Japan, was one of the first to evaluate the effect of SGLT2 inhibitors on vascular function [29]. This randomized controlled trial (RCT) involved 80 patients with T2DM and found that dapagliflozin 5 mg daily for 16 weeks significantly improved flow-mediated dilation (FMD), a well-recognized marker of endothelial function [29], by 1.05% as compared to a 0.94% reduction in FMD in the metformin group (*p* = 0.04) for a subset of patients with baseline glycated haemoglobin (HbA1c) greater or equal to 7% [29]. The beneficial effect of SGLT2 inhibitors on FMD was summarized in a meta-analysis. Using dapagliflozin significantly increased FMD by 1.66% (95% CI, 0.56–2.76) [20]. The improvement in FMD was confirmed in another meta-analysis [30]. When compared to sulphonylureas, SGLT2 inhibition was associated with a higher FMD by a mean difference of 1.89% (95% CI, 0.1–3.75) [31]. Other SGLT2 inhibitors, including tofogiflozin and iprafliglozin, also improved FMD [32,33]. Additionally, the effect of SGLT2 inhibitors has been evaluated using the EndoPAT technology. SGLT2 inhibition increased the reactive hyperaemia index, which is a marker of peripheral microvascular function, from 0.45 to 0.66 (*p* < 0.01) only in the dapagliflozin group [34]. 

Apart from peripheral endothelial function, the coronary endothelial function is another key vascular bed that could impact cardiovascular outcomes, especially heart failure. Dysfunctional endothelium in the coronary microcirculation could impact cardiomyocyte health, and the resultant coronary microvascular dysfunction is a key determinant in HFpEF [35]. The improvement in coronary endothelial function by SGLT2 inhibitors was demonstrated in preclinical models. Empagliflozin and dapagliflozin improved cardiac microvascular endothelial cell function in mice exposed to ischaemic reperfusion injury, with a resultant increase in endothelium-dependent relaxation and reduction in reactive oxygen species and endothelial cell activation markers [36,37]. In addition, in mouse cardiomyocytes, SGLT2 inhibitors appear to demonstrate a class effect in which the Na^+^/H^+^ exchanger is inhibited, lowering Na flux into cardiomyocytes and promoting coronary vasodilation [38].

Endothelial dysfunction contributes to arterial stiffness, a pathophysiological process involving remodeling the elastic fibers in arteries and fibrosis of arterial walls [28]. Arterial stiffness can lead to atherosclerosis through increased shear stress, stimulating collagen deposition and extracellular matrix deposition in the arterial wall and accelerating the formation of atherosclerotic plaques [28]. Arterial stiffness can be measured using pulse wave velocity (PWV), which indicates the transit time of the arterial pressure waves in the large arteries, which increases in stiff arteries. Increased wall pressure could aggravate endothelial dysfunction, leading to a vicious cycle of endothelial dysfunction and arterial stiffening [28]. A 1 meter/second (m/s) increase in PWV increases the risk of cardiovascular events by 7% [39]. SGLT2 inhibition has been shown to improve arterial stiffness. A meta-analysis comparing the PWV between the SGLT2 inhibitors—dapagliflozin and empagliflozin—versus placebo found that SGLT2 inhibitors reduced PWV by 0.76 m/s [40], while an observational study with tofogliflozin showed a worsening of arterial stiffness in non-tofogliflozin but not in patients on tofogliflozin [41]. Further details of the impact of SGLT2 inhibitors on vascular function measurements are presented in Appendix A.

### 3.2. Effects of SGLT2 Inhibitors on Mechanisms in Atherosclerosis

SGLT2 inhibitors reduce vascular inflammation.

The amelioration of vascular inflammation by SGLT2 inhibitors has been demonstrated in animal studies. Dapagliflozin reduces inflammatory cytokines, including tumor necrosis factor-alpha (TNF-α), endothelial adhesion molecules and nuclear factor-kappa B (NF-kB) expression in mice [42]. Similar findings have also been observed in visceral adipose tissue of Zucker diabetic fatty rats, in which IL-1β, IL-6, TNF-a, and monocyte chemoattractant (MCP)-1 significantly reduced after 6 weeks of empagliflozin administration [43]. SGLT2 inhibition modulates NLRP3 inflammasome activity in humans, reducing IL-1β secretion [23,44,45,46]. Regarding the inflammatory cytokine, IL-6, a recent meta-analysis involving more than 5000 adult individuals with T2DM, the use of SGLT2 inhibitor reduced IL-6 by a mean of 1.04 and 1.3 compared to other antidiabetic medications [21]. Dapagliflozin use was also associated with a reduction in TNF-α [46]. However, the impact of SGLT2 inhibition on CRP is not clear. While Wang et al. found that CRP level was reduced by SGLT2 inhibitors [47], Buttice et al. did not find improvements in CRP or TNF-α in a meta-analysis comprising 38 RCTs investigating the effect of SGLT2 inhibition in inflammatory markers [48]. 

2.SGLT2 inhibitors attenuate oxidative stress and atherosclerosis

Oxidative stress plays a significant role in the atherosclerotic process and is closely associated with endothelial dysfunction. For instance, the endothelial adhesion molecule, vascular cell adhesion molecule 1 (VCAM-1), activates nicotinamide adenine dinucleotide phosphate (NADPH) oxidase 2, generating reactive oxygen species (ROS) [49] and rapidly induces matrix metalloproteinases (MMPs) [50]. The MMPs subsequently disrupt cell-to-cell adhesions and extracellular matrix [50], separating the intima from the blood vessel’s medial lining, allowing smooth muscle cell migration and promoting atherosclerotic plaque formation [51]. Sustained exposure to excess ROS in the blood vessels, induced by mitochondria dysfunction, uncoupled nitric oxide synthase, myeloperoxidase, and NADPH oxidases [52], can induce endothelial dysfunction and drive inflammation further. SGLT2 inhibition has been shown to reduce leukocyte mitochondrial superoxide production [23,53], reduce NADPH and myeloperoxidase and increase the production of endogenous antioxidant glutathione [54]. The DEFENCE trial showed that 16 weeks of dapagliflozin reduced urine 8-hydroxy-2′-deoxyguanosine (8-OHdG), a marker of oxidative stress [29]. At the same time, studies on empagliflozin found increased mitochondrial superoxide dismutase activity, a key player in clearing ROS and reducing oxidative stress [23,53,55].

3.SGLT2 inhibitors reduce thrombosis

Platelet activation is a key process in promoting vascular inflammation through a sequential process involving rolling and firm adhesion of activated platelets to the endothelial surface [56,57,58]. These processes are mediated by specific receptors, such as P-selectin, and accelerate the recruitment of immune cells to the inflamed endothelium and subendothelial spaces to drive atherosclerosis [59]. Studies have shown that SGLT2 inhibitors reduce CD62-P antigen expression [23,60], a platelet activation marker stored in platelet granules that is rapidly mobilized to the cell surface upon activation [61]. This finding suggests a potential benefit of SGLT2 inhibitor in attenuating platelet activation, aggregation and atherothrombosis [62]. 

Appendix A presents further information about selected studies evaluating the impact of SGLT2 inhibitors on specific biomarkers.

### 3.3. SGLT2 Inhibitors and Heart Failure

Clinical trials and cardiac markers

SGLT2 inhibitors have been consistently proven to reduce heart failure-related events. The hard clinical outcomes are supported by strong biochemical evidence demonstrating a significant reduction in N-terminal pro b-type natriuretic peptide (NT-proBNP), a marker of ventricular wall stress, cardiac remodelling and dysfunction [63]. SGLT2 inhibitors, dapagliflozin and canagliflozin, reduce NT-proBNP between 11 to 18.2% [24,25], with the reduction sustained for up to 6 years [64] (Appendix A). Postulated mechanisms of SGLT2 inhibitors include beneficial haemodynamic effects through diuresis and natriuresis, resulting in reduced cardiac preload, afterload [65] and reverse ventricular remodelling [66]. Furthermore, SGLT2 inhibitors may attenuate myocardial injury and subclinical cardiac damage [67], as evidenced by a significant reduction in high-sensitivity troponin I (hs-TnI) by empagliflozin [24] (Appendix A). Additionally, SGLT2 inhibition may provide direct cardioprotection through metabolic reprogramming, which shifts energy utilization from fatty acid and carbohydrates toward ketone body oxidation in myocardial cells, resulting in greater efficiency in generating energy in the myocardium [68]. Improvement in coronary microvascular endothelial function could also explain the beneficial effects of SGLT2 inhibition, specifically in HFpEF. In this first study comprising 16 patients, 4 weeks of dapagliflozin improved myocardial flow reserve, a marker of coronary microcirculation and vasodilation [69]. In other preclinical models, SGLT2 inhibitors reduced cardiac fibrosis by suppressing endothelial-to-mesenchymal transition by inhibiting TGF-beta/Smad signalling [70]. These processes, along with other mechanisms, such as reducing inflammation and oxidative stress, work synergistically to contribute to the beneficial effects of SGLT2 inhibition on the myocardium. 

### 3.4. SGLT2 Inhibitors and the Kidneys

Clinical trials focusing on renal outcomes

The EMPA-KIDNEY trial, which involved patients with diabetic kidney disease, glomerular disease and hypertensive or renovascular disease, found that empagliflozin reduced kidney disease progression by 29% (HR 0.71; 95% CI, 0.62–0.81), with a 50% reduction in the rate of eGFR decline [71]. The CREDENCE trial recruited individuals with T2DM with or without CVD, but all had chronic kidney disease on stable maximally tolerated doses of an angiotensin-converting enzyme inhibitor or angiotensin receptor blocker. MACE was reduced by 32% (HR 0.68; 95% CI, 0.49–0.94) and 15% (HR 0.85; 95% CI, 0.69–1.06) in the primary and secondary prevention groups, respectively [13]. Canagliflozin was also associated with a relative risk reduction of 40% in doubling serum creatinine and a 32% reduction in reaching end-stage kidney disease [13]. The DECLARE-TIMI trial assigned patients with T2DM with or without CVD to dapagliflozin. Although dapagliflozin did not significantly lower MACE, there was a 17% reduction (HR 0.83; 95% CI, 0.73–0.95) in heart failure events [14]. On the other hand, the DAPA-CKD trial was terminated early due to efficacy. Dapagliflozin reduced the relative risk of sustained decline in renal function, end-stage kidney disease, or renal-related mortality by 39% (HR 0.61; 95% CI, 0.51–0.72) [72]. Figure 3 below summarizes the key renal outcome trials with SGLT2 inhibitors.

2.Mechanisms of SGLT2 inhibitors in renal protection.

Multiple pathophysiological processes contribute to the development of diabetic kidney disease (Figure 4). Oxidative stress, inflammation, and the increased release of growth-forming factors lead to podocyte damage, tubulointerstitial fibrosis and glomerulosclerosis [73]. These mechanisms, together with reduced sodium delivery to the glomerulus and altered tubuloglomerular feedback, induce glomerular efferent arteriolar vasoconstriction, increase intraglomerular pressure, and promote glomerulosclerosis [73]. SGLT2 inhibition blocks the reabsorption of sodium and glucose in the proximal tubules, restores sodium delivery to the macula densa, reverses efferent arteriolar vasoconstriction and relieves intraglomerular hypertension [73]. The benefits of SGTL2 inhibition have been validated through animal models and human studies that specifically measured changes in the glomerular and renal tubular injury markers (Appendix A). For example, SGLT2 inhibition has been shown to reduce urinary nephrin, a protein expressed in podocytes and a sensitive marker of podocyte injury [74]. Evidence suggests that increased urinary nephrin can be detected before albuminuria and is correlated with the severity of glomerular injury [75]. In addition, SGLT2 inhibition significantly increases transforming growth factor beta-1 (TGF-β) excretion [74]. TGF-β plays a crucial role in inducing hypertrophic response and is a key marker of renal fibrosis [76]. Another marker of renal fibrosis, MMP-7, is reduced by SGLT2 inhibition over 2 years [77]. Other markers studied include neutrophil gelatinase-associated lipocalin (NGAL), which is produced by neutrophils and released by injured renal tubular cells [78]. NGAL is significantly reduced by tofogliflozin, an SGLT2 inhibitor developed in Japan [79]. Another plasma biomarker is the liver-type fatty acid binding protein (L-FABP), secreted by renal proximal tubular cells in response to tissue hypoxia [80]. Canagliflozin use for 52 weeks has been shown to reduce urinary L-FABP significantly in T2DM individuals compared to non-users [81]. In terms of inflammatory markers, a post hoc analysis of the CANVAS trial showed a significant reduction in TNF receptors 1 and 2 [82], while other RCTs showed a reduction in IL-6 [21] and a proximal tubular injury marker [83], kidney injury molecule 1 [84,85] 

### 3.5. Adverse Effects Associated with SGLT2 Inhibitors

Particular caution needs to be taken when initiating SGLT2 inhibitors. Firstly, increased glucosuria predisposes to genitourinary infections. Meta-analyses reported a 2.9- to 3.3-fold increased risk for urogenital infections with SGLT2 inhibitors use in patients with T2DM [86]. Furthermore, the highest risk of genitourinary tract infections was reported to occur approximately in the first 28 days of initiation [87]. Fournier’s gangrene, a rare but potentially fatal perineal soft tissue infection, was found to be associated with SGLT2 inhibitors [86]. Secondly, the risk of hypotension from volume depletion, especially with concomitant use of diuretics, needs to be monitored closely, especially in susceptible patients who have borderline low blood pressure at the time of initiation [88]. Thirdly, significant glucose-lowering with SGLT2 inhibitors reduces insulin and increases glucagon secretion. This altered insulin and glucagon balance promotes free fatty metabolism and ketone generation. The enhanced risk of euglycemic diabetic ketoacidosis (DKA) needs to be emphasized, especially during periods of acute illness, poor appetite or significant stress (e.g., undergoing invasive procedures), when it is prudent to withhold SGLT2 inhibitors temporarily [89].

### 3.6. Dual SGLT1 and SGLT2 Inhibitors

There is emerging interest in combining SGLT-1 and SGLT-2 inhibition for potential synergistic effects. Sotagliflozin is the first FDA-approved dual inhibitor of SGLT-1 and SGLT2. SGLT2 is expressed mainly in the early segments of the proximal tubule of the nephron and mediates the reabsorption of more than 90% of filtered glucose [90]. In contrast, SGLT1 is expressed predominantly in the epithelial cells of the brush border in the small intestine [90], mediating intestinal glucose reabsorption. SGLT1 is also located in the late proximal tubule in the nephron, where it reabsorbs 3% of filtered glucose in normoglycemic individuals [90]. However, in hyperglycemia, the amount of glucose reabsorbed by SGLT1 in the proximal tubule increases significantly, hence supporting the role of dual SGLT1 and SGLT2 inhibition (Figure 5) [91]. In the intestines, SGLT1 inhibition inhibits intestinal glucose absorption and increases glucose delivery to the distal gut, stimulating GLP-1 secretion and further reducing postprandial hyperglycemia [92]. 

The main benefit of dual SGLT1 and SGLT2 inhibition is that it reduces heart failure events. The SCORED trial, which looked at the effects of sotagliflozin in more than 10,000 individuals with T2DM, chronic kidney disease and additional cardiovascular risk factors, demonstrated a reduction in the composite of cardiovascular mortality, heart failure hospitalizations and urgent visits by 26% (hazard ratio 0.74; 95% CI, 0.63–0.88) [93]. The SOLOIST-WHF trial, which recruited 1222 patients with T2DM who were recently hospitalized for decompensated heart failure, sotagliflozin users had reduced cardiovascular mortality and subsequent heart failure exacerbations by 33% (hazard ratio 0.67, 95% CI 0.52–0.85) [94] (Figure 1). 

While SGLT2 inhibitors did not show a significant effect on stroke reduction, dual SGLT1 and SGLT2 inhibition was associated with a significant decrease in all-cause stroke by 34% (hazard ratio 0.66; 95% CI 0.48–0.91) and ischaemic stroke by 32% (hazard ratio 0.68; 95% CI 0.47 to 0.99) [95]. 

The incremental benefit of SGLT1 inhibition in neuroprotection could be attributable to a few postulated mechanisms. Firstly, SGLT1 is expressed in the brain and is increased in patients with T2DM. SGLT1 inhibition may reduce blood–brain barrier permeability and cerebral injury [96]. In the gut, SGLT1 inhibition increases glucose delivery to the distal intestine, alters the intestinal microbiome by increasing short-chain fatty acids and stimulating GLP-1 secretion, which has anti-atherogenic properties, reduces inflammation, thrombosis and oxidative stress, and stabilizes atherosclerotic plaques [97]. 

## 4. GLP-1 Receptor Agonist (GLP-1 RA)

The GLP-1 RA is another class of antidiabetic agent with cardiovascular and renal benefits. GLP-1 is an incretin hormone secreted from enteroendocrine L cells with pleiotropic effects on glucose regulation [98]. GLP-1 stimulates insulin secretion, inhibits glucagon release, promotes satiety and reduces gastric emptying [99]. In T2DM, the incretin effect is diminished but can be restored by GLP-1 infusion [100], making GLP-1 mimetics attractive agents to improve glycemic control. Moreover, GLP-1 affects systemic homeostasis by activating receptors in various organs, including the kidneys, heart, adipose tissue, liver and muscles [101]. GLP-1 receptors in the brain regulate food intake directly and indirectly through vagal afferents in the gut [102].

The first GLP-1 RA, exenatide, was approved for clinical use in 2004. Exenatide was derived from exendin, an endogenous GLP-1-like peptide secreted in the saliva of the lizard Heloderma suspectum (Gila monster) [103]. Endogenous GLP-1 has a short half-life of 2 min due to rapid degradation by DPP-4 [104]. With modifications to the amino acid structure, the half-life of exenatide is prolonged to 2.5 h [103]. Retaining more than 50% homology as endogenous GLP-1, exenatide enhances pancreatic beta-cell secretion and reduces postprandial hyperglycaemia [103]. Lixisenatide, another GLP-1 RA designed based on exendin-4, is modified at different amino acids, prolonging its half-life by up to 4 h [105]. Other GLP-1 RAs, specifically liraglutide and semaglutide, involved creating conjugates with long-chain fatty acid moieties to render the molecular size of the GLP-1 RAs too large to be filtered by the kidneys effectively, thereby reducing renal clearance and prolonging the half-life [106]. 

Currently, the most commonly used GLP-1 RAs for treating T2DM are liraglutide, semaglutide and dulaglutide. In addition to T2DM, liraglutide and semaglutide are also approved for treating obesity without T2DM. High-dose liraglutide (3 mg/day) was evaluated in the SCALE trial, a 56-week study involving more than 3700 patients with BMI ≥ 30 or ≥27 with at least one obesity-related complication, found a greater mean weight of 5.6 kg in the high dose liraglutide group compared to placebo [107]. The STEP trial, which evaluated the efficacy of semaglutide in weight loss, found a mean weight loss of nearly 15% at week 68 compared to 2.4% for placebo among individuals with obesity but without T2DM [108]. 

In terms of cardiovascular risk reduction, there appears to be a difference between the older formulations (exenatide, lisixenatide) and the newer generations (liraglutide, dulaglutide and semaglutide). Exenatide was not superior to placebo in reducing cardiovascular outcomes in patients with T2DM [109]. This was similar to lixisenatide, in which the rates of cardiovascular events were not significantly different from placebo among patients with T2DM and recent acute coronary syndrome [110]. However, liraglutide was the first GLP-1 RA to demonstrate positive cardiovascular benefits [7]. The LEADER trial, published in 2016, randomized more than 9300 patients with T2DM and high cardiovascular risk to liraglutide or placebo. Liraglutide was associated with a 13% relative risk reduction (HR 0.87; 95% CI, 0.78–0.97) in three-point MACE over 3.8 years of follow-up [7]. Following the LEADER trial, the SUSTAIN-6 trial [111], which assessed semaglutide, and the REWIND trial [112], which evaluated dulaglutide, both reported a significant reduction in major cardiovascular events in patients with type 2 DM at high cardiovascular risk. A meta-analysis of eight CVOTs reported a 14% reduction (HR 0.86; 95% CI, 0.80–0.93) in MACE, with separation of event curves approximately 12–18 months after initiation [113]. Figure 6 summarizes the key cardiovascular outcome trials for GLP-1 RAs.

GLP-1 RAs confer cardiovascular benefits through the following mechanisms: they improve vascular function [31,40], reduce inflammation [114,115], oxidative stress [11], thrombosis [115] and have direct effects on the heart. Figure 7 summarizes the cardiovascular benefits of GLP-1 RAs.

### 4.1. Effects of GLP-1 RA on Endothelial Function and Arterial Stiffness

A network meta-analysis comparing the efficacies of various antidiabetic agents found that GLP1-RAs increased FMD by a mean difference of 3.70 (95% CI 1.39–5.97) compared to lifestyle intervention [31]. In comparison with sulphonylureas and placebo, GLP-1 RAs also showed a significant improvement in FMD of 3.33 (95% CI 1.36–5.34) and 3.30 (95% CI 1.21–5.43), respectively [31]. The physiological improvement in FMD was supported by mechanistic insights revealing an improvement in endothelial function in human umbilical vein endothelial cells via the mTOR/Akt signaling pathway, promoting endothelial NOS activity and increasing NO production [116]. In addition, improvement in endothelial function could also be attributed to an increase in the regenerative capacity of the endothelium, as demonstrated by an increase in endothelial progenitor cells after 12 weeks of treatment with dulaglutide [117]. 

Regarding arterial stiffness, a network meta-analysis comparing different classes of antidiabetic agents revealed that only GLP-1 RAs improve arterial stiffness by reducing pulse wave velocity by a mean difference of 1.06 (95% CI −2.05 to −0.10) in individuals with abnormal glucose metabolism [40]. This improvement in arterial stiffness could be supported by positive changes in the extracellular matrix in the blood vessels, as demonstrated by an experimental study evaluating the effects of semaglutide on vascular structure in high-fat diet-induced obese mice, which revealed decreased levels of ECM proteins Coll5a1, Lama4 and Sparc [118]. 

The effects of GLP-1 RAs on endothelial function and arterial stiffness are summarized in Appendix A.

### 4.2. Mechanisms of Cardiovascular Protection by GLP-1 RA

Inflammation

Attenuating inflammation is a key protective mechanism of GLP-1 RAs. Contrary to SGLT2 inhibitors, GLP-1 RAs are associated with a stronger signal in reducing inflammation. A meta-analysis showed significant reductions in CRP and TNF-alpha by GLP-1 RAs compared to standard antidiabetic agents or placebo, with a mean difference of −0.54 (95% CI, −0.75 to −0.34) and −0.39 (95% CI, −0.62 to −0.15) respectively [114]. Furthermore, a 180-day treatment with semaglutide or dulaglutide reduced IL-1β, suggesting possible beneficial effects on targeting the NLRP3 inflammasome pathway [119]. Liraglutide also reduces MCP-1 [115], which promotes the recruitment of monocytes and macrophages to the subendothelial layer [120]. VCAM-1, the endothelial cell surface adhesion that promotes migration across vessel walls [121], is significantly reduced by liraglutide [117,122]. Various evidence in preclinical models has also demonstrated the effects of GLP-1 RA in stabilizing atherosclerotic plaques through anti-inflammatory and anti-atherogenic mechanisms on endothelial cells, vascular smooth muscle cells, monocytes and macrophages [113]. 

2.Oxidative stress

Another cardioprotective mechanism of GLP1-RAs includes ameliorating oxidative stress, as demonstrated by a reduction in oxidative stress markers 8-OH-dG and 8-iso-PGF2a by liraglutide within 8 weeks [123]. Malondialdehyde (MDA), a compound derived from lipid peroxidation and a common biomarker to measure oxidative stress [104], was reduced by 0.84 (95% CI −1.61 to −0.06) compared to controls [124]. 

3.Thrombosis

GLP-1 RAs modulate thrombosis and fibrinolysis. In an RCT comparing liraglutide and sitagliptin (a DPP-4 inhibitor), plasminogen activator inhibitor 1 (PAI-1), an important player that inhibits fibrin clearance [115], was significantly reduced by liraglutide [115]. The reduction in PAI-I was observed as early as 2 weeks after initiating liraglutide and prior to weight loss [115]. In addition, it has been postulated that GLP-1 RAs may inhibit intra-arterial thrombus formation in mice by inhibiting platelet reactivity [125]. An RCT evaluating the antiplatelet effect of GLP-1 RAs found that liraglutide reduces thromboxane-dependent platelet activation, as reflected by the reduction in urinary excretion of U-11-dehydro-TXB_2_, a metabolite of thromboxane A2 [126]. As GLP-1 receptors are widely distributed in the central nervous system and GLP-1 RAs can cross the blood–brain barrier, the aforementioned benefits of GLP-1 RAs could have significant neuroprotective effects [127] and could account for the 24% relative risk reduction in nonfatal stroke (HR 0.76; 95% CI, 0.61–0.95), a prespecified secondary outcome in the REWIND trial (dulaglutide) [128], and for the 39% (HR 0.61; 95% CI, 0.38–0.99) for nonfatal stroke in the SUSTAIN-6 trial (semaglutide) [111]. 

### 4.3. Effects of GLP-1 RA on Heart Failure

Emerging evidence supports the potential role of GLP-1 RA in heart failure (Figure 6). In a post hoc analysis of the REWIND trial, dulaglutide use was associated with a slower rise in NT-proBNP [128]. In the LIVE study, 241 individuals with HFrEF were randomized to liraglutide versus placebo for 24 weeks. In patients with T2DM, NT-proBNP decreased by 25% in the liraglutide group compared with placebo [129]. Liraglutide also reduced NT-proBNP significantly compared to metformin [130]. In the recent STEP-HFpEF trial looking at the effect of semaglutide in obese patients with HFpEF, semaglutide was associated with significant improvement in physical symptoms and exercise capacity, as well as a significant reduction in NT-proBNP by 25% [131]. The beneficial effects of GLP-1 RAs on HFpEF could be related to the alteration of epicardial adipose tissue (EAT) [132]. Evidence suggests that individuals with diastolic dysfunction have increased epicardial adipose tissue, which could lower peak oxygen consumption and worsen exercise tolerance [133]. Using liraglutide over 6 months, EAT thickness was significantly reduced from 9.6 to 6.26 mm versus no change in the metformin arm [134]. A reduction in EAT could attenuate left ventricular remodeling. Furthermore, reducing vascular inflammation, endothelial dysfunction, and oxidative stress could improve myocardial function [131]. Additionally, GLP-1 RAs inhibit the sodium-hydrogen exchanger 3 in proximal tubular cells to increase natriuresis and diuresis [135], contributing to favorable haemodynamic effects in heart failure. A summary of the impact of cardiovascular protection by GLP-1 RAs is presented in Appendix A. 

### 4.4. Effects of GLP-1 RA on Renal Function

GLP-1 RAs reduced composite renal outcomes (development of macroalbuminuria, decline in renal function, renal replacement therapy or renal-related mortality) by 21% [136]. The mechanisms underpinning the benefits of GLP-1 RAs on renal outcomes could be attributed to the presence of GLP-1 receptors in the renal glomeruli and mesangial cells, attenuating the inflammatory response to advanced glycation end products and reducing the progression of diabetic kidney disease [137]. In addition, GLP-1 RAs reduce oxidative stress by upregulating cyclic adenosine monophosphate (cAMP) and increasing protein kinase A, reducing the apoptosis of kidney podocytes and slowing down renal glomerulosclerosis [138] (Figure 6). The activation of GLP-1 receptors in the kidneys leads to direct renal benefits. These include increased natriuresis, the reduction of vasoconstrictors such as angiotensin II and endothelin-1, and enhanced endothelium-dependent vasodilation [100]. The FLOW trial, a dedicated renal outcome trial that aimed to evaluate the efficacy of semaglutide in slowing the progression of renal impairment in individuals with T2DM and CKD, was prematurely terminated in October 2023 for efficacy [139]. The results were highly favorable. Among the 3533 participants with T2DM and CKD who were followed up for a median of 3.4 years, the risk of composite onset of end-stage renal disease—at least a 50% reduction in renal function or renal-related mortality—was reduced by 24% in the semaglutide group compared with placebo (HR 0.76; 95% CI 0.66–0.88) [140]. The SOUL trial is investigating the efficacy of oral semaglutide on combined cardiovascular and renal outcomes in 9650 individuals with T2DM [141]. The REMODEL trial, an ongoing mechanistic study investigating the effects of semaglutide on kidney oxygenation, inflammation and fibrosis, is planned to be completed by the end of 2024 (NCT04865770) (Figure 5) [142]. 

### 4.5. Adverse Effects Associated with GLP-1 RA

The most common adverse effect associated with GLP-1 RA is gastrointestinal intolerance, which tends to occur within one month of initiation [143]. More serious adverse effects include acute pancreatitis, which was reported to occur in approximately 3% of users [144]. The association between GLP-1 RA and thyroid cancer is unclear. While a nested case-control study reported a possible increased risk for all thyroid cancer by nearly 1.6 times and medullary thyroid cancer by 1.8 times [145], a recent cohort study did not show this association [146]. Regarding pancreatic cancer, GLP-1 RA is postulated to induce the proliferation of pancreatic islet cells and may stimulate pancreatic carcinogenesis in susceptible individuals [147]. However, there is a growing body of literature that suggests the lack of association between GLP-1 RA and pancreatic cancer [148]. With the increasing use of GLP-1 RA, long-term follow-up of data will provide more definitive evidence regarding cancer risks with GLP-1 RA.

### 4.6. Combination Agents with GLP-1

An emerging field in the pharmacological landscape for T2DM involves combining GLP-1 RAs with other agents. These include amylin analogue, GIP receptor agonist and glucagon receptor agonist.

GLP-1 + amylin analogue

One of these combinatory approaches involves a long-acting amylin analogue, cagrilintide, and semaglutide (CagriSema). This agent has recently shown more significant weight loss than the individual components. In a small exploratory trial involving 92 participants randomly assigned to CagriSema, semaglutide or cagrilintide, the mean change in HbA1c from baseline to week 32 was 2.2% for Cagrisema and weight loss of 15.6% for CagriSema versus 5.1% for semaglutide alone [149]. It has been postulated that amylin agonists could delay gastric emptying, reduce postprandial glucagon secretion, and improve insulin sensitivity synergistically when added to GLP1 receptor agonists [149]. 

2.GLP-1 + GIP receptor agonist (tirzepatide)

Unimolecular dual agonists have emerged as a promising strategy to enhance the metabolic effects of GLP-1 receptor agonists by combining them with other enteropancreatic hormones. An example of such a dual agonist agent is tirzepatide (LY3298176), which combines the agonist activities of GLP-1 and GIP receptors [150]. Tirzepatide has a greater affinity to GIP receptors and can be administered once weekly [150]. 

Tirzepatide has recently been extensively evaluated in phase 2 trials with either placebo (SURPASS-1) [151] or other active comparators, including semaglutide (SURPASS-2) [152], degludec (SURPASS-3) [153], glargine (SURPASS-4) [154] or add-on to basal insulin (SURPASS-5) [155]. At an incremental dose to a maximum of 15 mg per week, tirzepatide showed superiority in HbA1c and bodyweight reduction at 40 or 52 weeks. Ongoing studies are currently assessing the impact of switching GLP-1 RA to tirzepatide (SURPASS SWITCH: NCT05564039 and SURPASS SWITCH 2: NCT05706506). The use of tirzepatide in recently diagnosed T2DM is also being evaluated in the SURPASS-early trial (NCT05433584). Furthermore, the combination agent Cagrisema (cagrilintide and semaglutide) is compared against tirzepatide to determine the degree of HbA1c reduction (NCT062219690). 

Apart from T2DM, tirzepatide is also extensively studied in patients with obesity without T2DM. In the SURMOUNT-1 trial, which recruited more than 2500 subjects with obesity but without T2DM to tirzepatide versus placebo, 91% (95% CI, 88–94) of participants were able to achieve at least 5% of weight loss versus 35% (95% CI, 30–39) with placebo [156]. More than 50% of participants who received the highest dose of tirzepatide could lose more than 20% of their weight, compared to just 3% in the placebo group [156]. The safety profile of tirzepatide was similar to GLP-1 receptor agonists, with mild to moderate gastrointestinal symptoms being the predominant adverse effects, which improved over time [156]. 

The cardiovascular benefits of GIP could be directly associated with the anti-atherogenic effects of GIP on human umbilical vein endothelial cells [157]. GIP reduces gene expression of advanced glycation end-product receptors and other pro-atherogenic molecules such as VCAM and PAI-I [158]. GIP also activates endothelial nitric oxide synthase, which synthesizes nitric oxide [157], a key vasodilator inhibiting vascular smooth muscle cell proliferation [159]. In addition, active GIP has also been shown to suppress inflammatory responses in macrophage foam cells, reduce expression of CD36, and promote anti-inflammatory and anti-oxidative stress effects by upregulating the cAMP pathway [160]. A post hoc analysis of the SURMOUNT-1 trial also showed a possible preferential reduction in triglyceride-rich lipoprotein [161]. All these actions contribute to atherosclerotic plaque stabilization in animal models [162]. 

Regarding the cardiovascular benefits of tirzepatide in humans, a meta-analysis comprising 4887 participants treated with tirzepatide reported a hazard ratio of 0.80 for four-component major adverse cardiovascular events and 0.80 for all-cause mortality (95% CI, 0.57–1.11) [163]. In a posthoc analysis of SURMOUNT-1, which recruited adults who were obese but without T2DM, tirzepatide use was associated with a significant reduction in predicted 10-year cardiometabolic risk score [164]. In terms of individual risk factors, tirzepatide leads to greater weight loss, improves glycemic control, and reduces blood pressure by up to 11.5 mmHg with the highest dose [165]. The ongoing SURPASS-CVOT trial will provide insights into the cardiovascular outcomes of tirzepatide against dulaglutide over 54 months of follow-up [166], while the SUMMIT trial will inform about the utility of tirzepatide in heart failure (NCT04847557).

In terms of renoprotection, tirzepatide was evaluated in the SURPASS-4 trial, in which subjects with CVD were recruited. The risk of a composite renal outcome, including progression to ESRD, new-onset macroalbuminuria and/or deterioration in renal function or renal-related mortality, was significantly reduced by 42% (hazard ratio 0.58; 95% CI 0.43–0.80) compared to glargine alone [167]. The molecular mechanisms in renoprotection have been primarily attributable to the direct effects of GLP1 receptor activation and the indirect effects of GIP, including the reduction in proinflammatory cytokines, oxidative stress, weight loss and glycemic control [168]. A mechanistic study looking at changes in kidney oxygenation via MRI (TREASURE CKD trial: NCT04847557) is currently ongoing to compare tirzepatide versus placebo in patients who are overweight, have albuminuria and have chronic kidney disease regardless of glycemic status (TREASURE CKD; NCT04847557).

Appendix A provides more information about the series of completed and ongoing clinical trials involving tirzepatide among patients with or without T2DM. 

3.GLP-1 + glucagon receptor agonist

Glucagon, produced by the pancreatic alpha cells, is conventionally viewed as having directly opposite effects of insulin in glucose homeostasis [169]. Given the known association of glucagon with hyperglycemia, glucagon agonism may appear counterintuitive in treating T2DM and obesity. However, glucagon has been found to have pleiotropic metabolic actions [170]. In terms of lipid metabolism, hepatic glucagon receptor activation enhances hepatic lipid catabolism and facilitates the beta-oxidation of fatty acids, making this pathway an attractive therapeutic strategy in managing hepatic steatosis [171]. Other benefits of glucagon in energy metabolism include delaying gastric emptying and upregulating amino acid and fatty acid metabolism [172]. 

Cotadutide (MEDI0832), the first GLP-1/glucagon co-agonist tested in human clinical trials, showed promising results in weight loss and glucose-lowering [173]. A phase II trial in 2018 recruited 65 patients with obesity and T2DM, who were randomized to cotadutide versus placebo for 49 days [174]. The cotadutide arm was associated with a significant 21.5% reduction in glucose excursion post-mixed-meal tolerance test compared to 6.3% (*p* < 0.001) for the placebo arm [174]. The per cent body weight loss in the cotadutide arm was 3.4% compared to 0.08% for placebo (*p* = 0.002) [174]. In addition, there was a dose-dependent improvement in glucose and weight reduction, with efficacy already observed at the lowest dose level of 50 mcg with no signs of a plateau over the 49-day trial period [174]. Another GLP-1/glucagon agonist that has advanced to human clinical trials is efinopegdutide, which has a relative potency ratio of approximately 2:1 for GLP1 and glucagon receptor, respectively [175]. In a phase II trial comparing efinopegdutide with liraglutide 3.0 mg involving more than 340 participants, the weight loss achieved in the epfinopegdutide group was nearly twice compared to the liraglutide group (10% vs. 5.8%) [176]. However, almost 90% of participants taking epfinopegdutide experienced gastrointestinal adverse effects compared to 60% taking liraglutide [144]. Epfinopegdutide is currently undergoing repositioning as a once-weekly treatment for non-alcoholic fatty liver disease NAFLD [175]. The latest addition to the line of GLP-1/glucagon co-agonist is survodutide (BI 456906), which was recently tested in a phase II trial involving 387 obese participants without T2DM for 46 weeks [177]. The highest dose led to nearly 15% weight loss [177]. Another phase II trial to characterize the dose range and escalation involved 411 patients with T2DM and compared to semaglutide 1 mg weekly found that a dose-dependent absolute reduction in HbA1c up to nearly 1.7% after 16 weeks, as compared to approximately 1.5% for semaglutide [178]. 

4.GLP-1-glucagon agonism and potential cardiorenal benefits

In the vasculature, glucagon may have vasoactive effect, as demonstrated in animal models, via the activation of both the glucagon and GLP-1 receptors, demonstrating a possible crosstalk between the two hormone receptors [179]. It is currently unclear whether similar crosstalk is observed in humans, in which GLP-1 glucagon coactivation may lead to a greater degree of vasodilation than GLP-1 alone. Glucagon receptors are also located in the heart [180]. Pharmacological glucagon administration has been shown to lead to a transient increase in heart rate and cardiac output [180]. However, the effects of prolonged glucagon receptor activation in the myocardium are unclear. In the kidneys, glucagon receptors are located along the nephron and are crucial to the regulation of metabolic functions, oxidative stress, inflammation and fibrosis [181]. Data from a phase II clinical trial evaluating cotadutide in patients with T2DM and chronic kidney disease reported a 51% reduction in urinary albumin-to-creatinine ratio at day 32 compared to placebo. Even though this result was trending to significance (*p* = 0.0504) [182], the degree of albuminuria reduction was higher than the 18–26% reduction in albuminuria with the use of GLP-1 receptor agonist alone [182]. This is a promising result in renoprotection and warrants more extensive studies over a longer duration to clarify the superiority of combined GLP-1-glucagon agonist in nephropathy compared to GLP-1 RA agonist alone [182]. There is a lack of cardiovascular outcome data for GLP-1-glucagon agonist, and further studies are warranted to assess the long-term safety and efficacy of GLP-1-glucagon agonists. 

5.Triple GLP-1 + GIP + glucagon receptor agonist (retratrutide)

There is significant interest in using multi-receptor agonists in treating T2DM and obesity given their additive effects on glucose-lowering and weight reduction. A triple agonist could lead to greater weight loss than its single- or dual-agonist counterparts. Retratrutide (LY3437943), a novel triple GLP-1, GIP, and glucagon receptor agonist, has a greater insulinotropic effect from both GIP and GLP-1 receptors than the glucose intolerance induced by the glucagon receptor [183]. Promising results have emerged from preclinical and recent human clinical trials. In a recent phase 2 trial involving 222 participants with T2DM and a BMI between 25 and 50 kg/m^2^ who completed the trial, the administration of retratrutide was associated with more significant HbA1c reductions than placebo and dulaglutide in a dose-dependent manner [184]. In another phase II trial, involving 338 adults with a BMI of 30 kg/m^2^ and above or BMI 27–30 kg/m^2^ with at least one weight-related condition, weight reduction was nearly 24% in the highest dose of retratrutide group compared to the 2% weight loss observed with dulaglutide [185]. Retratruide was generally well tolerated, with mild-to-moderate gastrointestinal adverse events reported in 35% of users, similar to the dulaglutide group [184]. The ongoing phase 3 Triumph trial that aims to recruit 1800 participants with severe obesity and established cardiovascular disease will provide insights about the efficacy of retratrutide in weight loss in this group of high-risk individuals (NCT05882045) [186]. 

## 5. Novel Glucose-Lowering Agents in Early-Phase Trials

Glimin

Imeglimin is the first oral agent in the novel class of oral antidiabetic agents known as ‘glimins’ [187]. Despite the structural similarity to metformin, the glucose-lowering mechanisms differ [188]. Imeglimin primarily acts at the mitochondria, improving mitochondrial morphology and function [189]. Furthermore, imeglimin also reduces the apoptosis of beta-cells [190]. In mouse models, a single dose of imeglimin has been shown to enhance insulin secretion [190]. Having been tested in seven clinical trials, a meta-analysis conducted on 1454 patients with T2DM treated with imeglimin found a significant 0.85% reduction in HbA1c and improved insulin resistance measured through homeostasis model assessment of insulin resistance (HOMA-IR) by 0.46 [191]. The main side effects of imeglimin are gastrointestinal intolerance [192]. Imeglimin is currently approved for treating T2DM in Japan and India [193]. 

Beyond the glucose-lowering, imeglimin can potentially offer cardiovascular benefits. In animal models, imeglimin has been shown to improve cardiovascular function [194]. The underlying mechanisms can be attributable to improved nitric oxide synthesis, reduced oxidative stress specifically produced by reverse electron transport in the mitochondria, and the minimized hyperglycemia-induced apoptosis of endothelial cells [195]. An improvement in endothelial function due to imeglimin has been demonstrated by an improvement in postprandial FMD in humans [196]. Regarding cardiac function, dysfunctional unfolded protein response has been postulated to lead to heart failure with a preserved ejection fraction. Imeglimin ameliorated HFpEF by normalizing the protein response [197]. The data on renoprotection are currently limited to animal models. A 90-day course of imeglimin treatment has been shown to reduce interstitial fibrosis and albuminuria in rat models [198]. The further evaluation of glimins in larger patient cohorts with longer follow-up is required to establish the additional cardiovascular and renal benefits beyond glucose-lowering.

2.Dual peroxisome proliferator-activated receptor (PPARα/γ) agonist

PPARs belong to the nuclear hormone receptor family essential to lipid and glucose metabolism [199]. Dual PPAR agonists combine the insulin-sensitizing effects of PPAR-alpha activation with the lipid-modifying properties of PPAR-gamma, making this agent potentially useful in treating diabetic dyslipidaemia. Aliglitazar, a dual PPAR agonist, had previously shown promising results in glucose-lowering. In a pooled analysis of data from 3 RCTs comprising 591 subjects, aliglitazar use was associated with a 1.08% reduction in HbA1c compared to 0.22% with placebo [200]. However, the development of aliglitazar was terminated after the AleCardio phase III trial showed a lack of efficacy and increased risk of heart failure, renal dysfunction and gastrointestinal haemorrhage [201]. On the other hand, saroglitazar, a newer dual PPAR agonist, has been approved for use in India since 2013 [202] and Mexico [203] following successful clinical trials. In the PRESS trials, saroglitzar was shown to reduce triglyceride by up to 45%, along with improvements in glycemic control [204,205]. Saroglitazar may address residual cardiovascular risks associated with high non-HDL cholesterol [206] and small, dense LDL cholesterol commonly observed in diabetic dyslipidaemia [207]. Given the potential role of saroglitazar in NAFLD, a phase 4 study that aims to recruit 1500 subjects with NAFLD with comorbidities such as T2DM, dyslipidaemia or metabolic syndrome was commenced in India in 2023 (NCT05872269) [208]. Although not reported to cause cardiovascular complications, it would be prudent to avoid or use dual PPAR agonists cautiously in patients with a history of congestive cardiac failure [209]. 

Apart from PPAR dual agonist, a pan-PPAR agonist targeting PPAR alpha/delta/gamma has been developed and recently approved for use in patients with T2DM and non-alcoholic steatohepatitis in China [210]. In a phase III trial, chiglitazar significantly improved insulin sensitivity and HOMA-IR compared to placebo or sitagliptin over 24 weeks of treatment [211]. Regarding lipid profiles, chiglitazar reduced triglycerides but increased LDL-cholesterol after 24 weeks of treatment [212]. A phase 3 trial is ongoing to assess the efficacy and safety of chglitazar as an add-on to metformin in China (NCT04807348) [213]. 

3.Glucokinase activator

Glucokinase is an enzyme that functions as a glucose sensor for stimulating insulin secretion and reducing fasting and postprandial glucose [214]. In T2DM, glucokinase expression is significantly reduced, resulting in an attenuated glucose sensitivity [215]. Therefore, glucokinase activators (GKAs) represent a novel class of antihyperglycemic agents that can potentially stimulate insulin secretion and promote hepatic glucose uptake in response to varying glycemic levels [216]. Partial, hepatoselective glucokinase activation reduces hypoglycemia risk [216]. 

Dorzagliatin, a mixed hepatopancreatic GKA, was shown to reduce HbA1c by 1.12% in the 75 mg twice-daily group compared to placebo [217]. There were no serious adverse events, and the incidence of adverse events or hypoglycemia was similar to that of the placebo [217]. Dorzaglipatin has progressed to a phase III trial, which showed that patients receiving dorzagliptin and metformin had a 1.07% decrease in HbA1c, compared to 0.5% for placebo [218]. However, dorzagliatin was associated with a significant elevation of triglycerides and body weight compared to placebo [219]. Further evaluation in longer clinical trials with active comparators is warranted to determine the long-term utility of GKAs. 

PB-201 is a partial hepatopancreatic GKA that is investigated in clinical trials in China [220]. In a phase 1 trial, PB-201 100 mg twice daily reduced HbA1c by up to 1.9% [221]. In a phase 2 study, HbA1c levels reduction after 2 weeks of PB-201 was similar compared to sitagliptin [220]. PB-201 has progressed to phase 3, and an ongoing trial among 672 Chinese participants will aim to assess its safety and efficacy compared with vildagliptin and placebo over 24 weeks [220]. 

4.G-protein coupled receptor 40 (GPR40) agonist

The activation of GPR40 stimulates insulin secretion [222], making this a potential target for T2DM. Expressed in the pancreatic beta cells, the activation of GPR40 promotes glucose-dependent insulin secretion and ensures glucose concentration remains regulated in physiological concentrations while minimizing hypoglycemia [223]. Despite efficacy in postprandial glucose control comparable to glimepiride, earlier GPR 40 compounds such as fasiglifam (TAK-875) had been terminated due to idiosyncratic drug-induced liver injury [224]. It was postulated that fasiglifam inhibited biliary transporters, resulting in bile acid accumulation in the liver [225]. Furthermore, it was also postulated that mitochondrial dysfunction induced by the earlier GPR40 agents contributed to hepatic adverse effects [225]. 

CPL207280, the latest GPR40 agonist, was carefully designed to address potential safety concerns. In human hepatocyte cultures, CPL207280 had a negligible effect on hepatic mitochondria, and no deleterious hepatic effects were observed, suggesting the DILI observed with TAK-875 could be drug-specific and not a class-specific adverse effect [225]. In a phase 1 study, 14 days of CPL207280 administration was safe and well tolerated by healthy volunteers [226]. 

Figure 8 illustrates selected novel glucose-lowering agents. Appendix A provides information on recently completed [220,226,227,228,229,230,231,232] and ongoing early-phase clinical trials involving novel glucose-lowering agents.

## 6. Future Directions and Conclusions

Looking back at the past decade, the release of the EMPA-REG trial results was a pivotal point that heralded a paradigm shift in the management of T2DM. The focus of T2DM management has shifted from a glucose-centric approach to one that comprehensively mitigates cardiovascular and renal risks. Over the past decade, there have been significant advances in the treatment of T2DM, with several new antidiabetic agents demonstrating robust cardiorenal benefits. The indications for these agents have since expanded to treat heart failure and kidney disease in patients without T2DM. 

The management of T2DM is evolving rapidly, with more than 400 early-phase interventional studies registered on clinical trial platforms. In the coming decade, the focus will be on testing these novel agents via high-quality clinical trials with adequate participant retention and follow-up to assess their long-term efficacy. Trials that compare novel agents against placebo will be insufficient, and active comparators need to be carefully selected to meet the new standards of care in T2DM management. Intensive research into the synergistic effects of combination agents targeting different mechanisms in glycemic control and cardiorenal protection will yield further insight into the clinical efficacy and cost-effectiveness of such approaches.

## Figures and Tables

**Figure 1 biomedicines-12-01386-f001:**
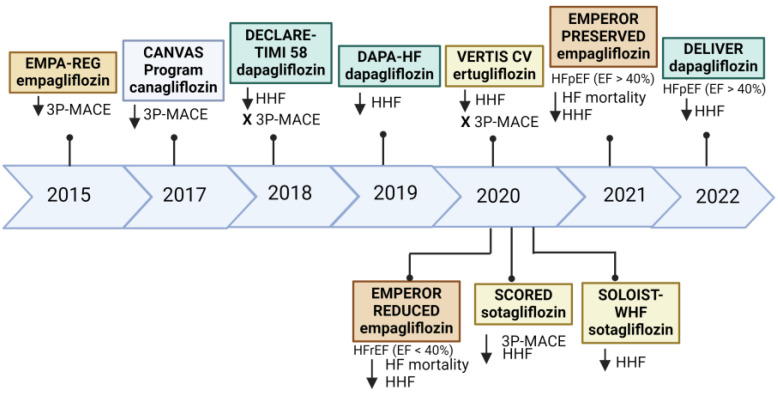
Key SGLT2 inhibitors cardiovascular outcome trials. Created with BioRender.com. 3P-MACE: 3-point major atherosclerotic cardiovascular events; HF: heart failure; HHF: hospitalization for heart failure; HFpEF: heart failure with preserved ejection fraction; HFrEF: heart failure with reduced ejection fraction.

**Figure 2 biomedicines-12-01386-f002:**
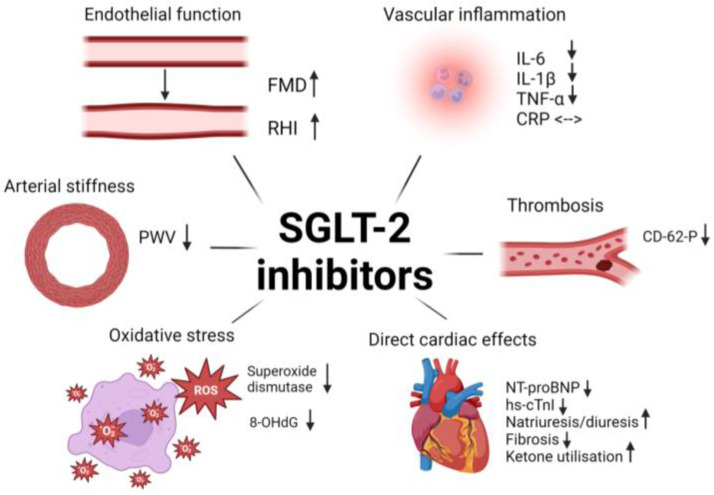
Summary of mechanisms of SGLT2 inhibitors in cardiovascular protection. Created with BioRender.com. FMD: flow-mediated dilation; RHI: reactive hyperaemia index; PWV: pulse wave velocity; IL: interleukin; TNR: tumor necrosis factor; 8-OHdG: 8-hydroxy-2′-deoxyguanosine; NT-proBNP: N-terminal pro b-type natriuretic peptide; hs-cTnI: high-sensitivity cardiac troponin T and I.

**Figure 3 biomedicines-12-01386-f003:**
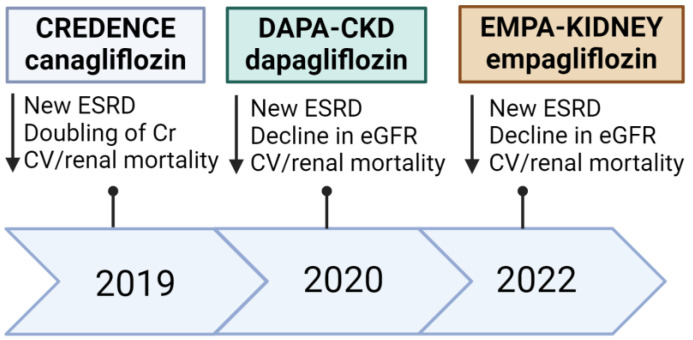
Key SGLT2 inhibitors renal outcome trials. Created with BioRender.com. ESRD: end stage renal disease; Cr: creatinine; CV: cardiovascular; eGFR: estimated glomerular filtration rate.

**Figure 4 biomedicines-12-01386-f004:**
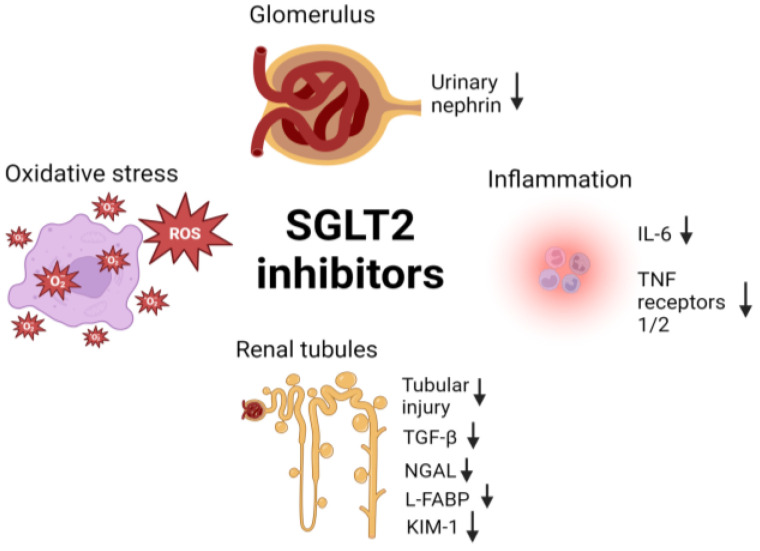
Mechanisms of renal protection by SGLT2 inhibitors. Created with BioRender.com. IL-6: interleukin-6; TNF: tumor necrosis factor: TGF-β: transforming growth factor β; NGAL: neutrophil gelatinase associated lipocalin; L-FABP: liver-type fatty acid binding protein; KIM-1: kidney injury molecule 1.

**Figure 5 biomedicines-12-01386-f005:**
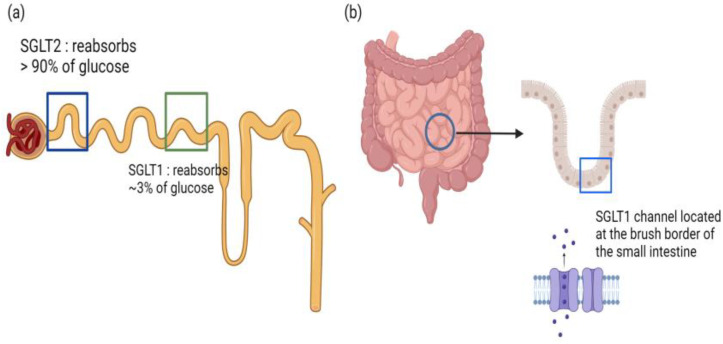
(**a**): Comparative differences in glucose reabsorption at proximal renal tubule between SGLT1 and SGLT2 receptors. (**b**): Main glucose reabsorption in the small intestine by SGLT1 receptors. Created with Biorender.com.

**Figure 6 biomedicines-12-01386-f006:**
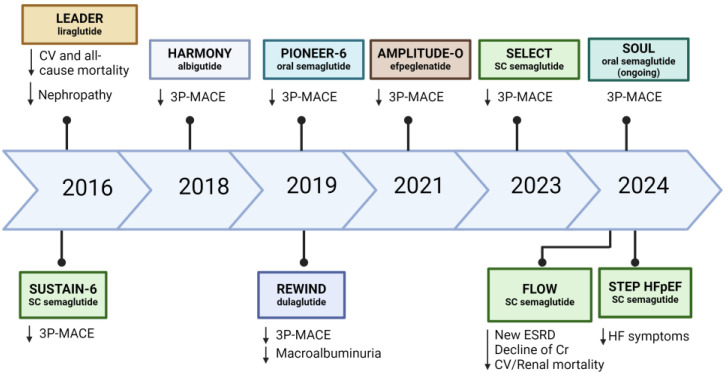
Key cardiovascular outcome trials for GLP-1 RAs. Created with BioRender.com. 3P-MACE: 3-point major atherosclerotic cardiovascular events; ESRD: end stage renal disease; CV: cardiovascular; HF: heart failure; HFpEF: heart failure with preserved ejection fraction.

**Figure 7 biomedicines-12-01386-f007:**
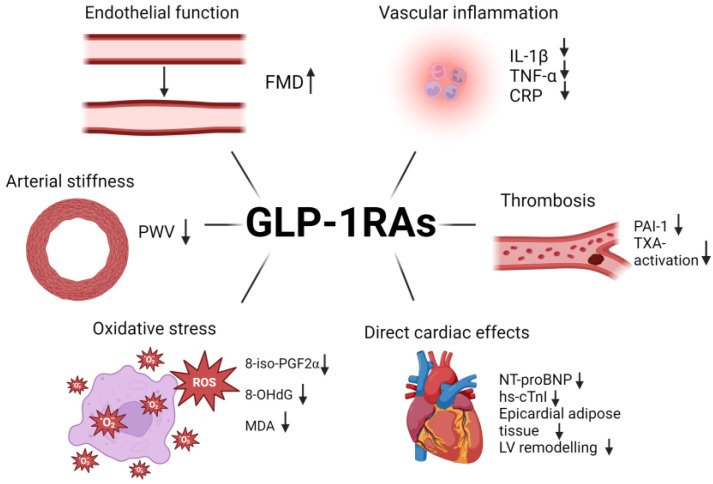
Mechanisms conferring cardiovascular protection by GLP-1 RAs. Created with BioRender.com. PWV: pulse wave velocity; PAI-1: plasminoagen activator inhibitor-1; TXA: thromboxane; TNF-α: tumor necrosis factor α; CRP: C-reactive protein; IL-1β: interleukin 1β; 8-iso-PGF2α: 8-iso-prostaglandin F2α; 8-OHdG: 8-hydroxy-2′-deoxyguanosine; MDA: malondialdehyde; hs-cTnI: high-sensitivity cardiac troponin I; NT-proBNP: N-terminal pro b-type natriuretic peptide; LV: left ventricle.

**Figure 8 biomedicines-12-01386-f008:**
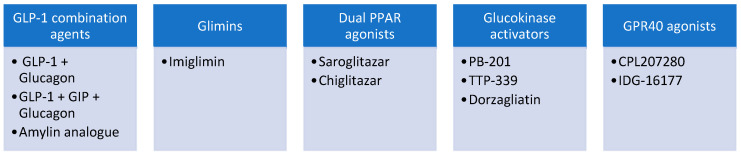
Novel glucose-lowering agents in early-phase clinical trials.

## Data Availability

No new data were created or analyzed in this study. Data sharing is not applicable to this article.

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
