# Peer review of "Novel Therapeutics for Type 2 Diabetes Mellitus—A Look at the Past Decade and a Glimpse into the Future"

_biomedicines, 2024, doi:10.3390/biomedicines12071386_

Round 1
Reviewer 1 Report
Comments and Suggestions for Authors
I enjoyed reading this interesting review. The manuscript is well structured, quite updated and well written. The topic is clinically very hot.
The figures are clear.
This review suggests only some additions to the text where the references are less up to date:
1- The paragraph on the endothelial protective effects of SGLT2i could be enriched. Indeed, the current experimental evidence mainly displays the antioxidant and anti-inflammatory actions of gliflozins, both at the macrovascular and microvascular level, including the coronary microvascular endothelium. The modulation of these and other pathways ultimately improves endothelial function, the first step involved in the development of atherosclerosis and the impairment of heart function, two key determinants of future CV events (Biomedicines 2021, 9(10), 1356; doi: 10.3390/biomedicines9101356). This issue should be better addressed by the authors.
2- A recent review has in-depth described the link between the pathophysiological mechanisms and clinical effects of SGLT2i on heart failure (Int J Mol Sci. 2021 May 30;22(11):5863. doi: 10.3390/ijms22115863). This topic could be integrated into the text.
Author Response
|
Thank you very much for taking the time to review this manuscript. Please find the detailed responses below and the revisions/corrections highlighted in the re-submitted files. |
|
Comment 1: This review suggests only some additions to the text where the references are less up to date:
The paragraph on the endothelial protective effects of SGLT2i could be enriched. Indeed, the current experimental evidence mainly displays the antioxidant and anti-inflammatory actions of gliflozins, both at the macrovascular and microvascular level, including the coronary microvascular endothelium. The modulation of these and other pathways ultimately improves endothelial function, the first step involved in the development of atherosclerosis and the impairment of heart function, two key determinants of future CV events (Biomedicines 2021, 9(10), 1356; doi: 10.3390/biomedicines9101356). This issue should be better addressed by the authors. |
|
Response 1: Thank you for highlighting the importance of SGLT2 inhibition on coronary microvascular endothelium and providing the references. We have added a paragraph (page 7, lines 18 to 29) to highlight evidence supporting the use of SGLT2 inhibitors to improve cardiac microvascular endothelial cell function.
|
|
Comment 2: A recent review has in-depth described the link between the pathophysiological mechanisms and clinical effects of SGLT2i on heart failure (Int J Mol Sci. 2021 May 30;22(11):5863. doi: 10.3390/ijms22115863). This topic could be integrated into the text |
|
Response 2: Thank you for the suggestion to integrate, in greater detail, the impact of SGLT2 inhibition on heart failure. We have added a short segment on page 9, lines 22 to 26 to incorporate the effect of SGLT2 inhibition on coronary endothelial function.
|
Reviewer 2 Report
Comments and Suggestions for Authors
Dear Authors,
You present here a review regarding the novel therapeutics for type 2 diabetes.
I suggest to detail the Introduction part, especially due to the fact that your manuscript wants to be a review.
Eliminate the spaces between paragraphs on pages 3 and 4. Reduce the figure dimension.
You summarized and presented well the advantages of the novel therapeutical agents. Also, you presented the substances that are in clinical studies and identified the future directions in the treatment of diabetes mellitus.
The references are well chosen and in agreement with the subject presented.
Author Response
|
Thank you very much for taking the time to review this manuscript. Please find the detailed responses below and the revisions/corrections highlighted in the re-submitted files. Comment 1 - I suggest to detail the Introduction part, especially due to the fact that your manuscript wants to be a review. Response 1 – Thank you for the suggestion to detail the Introduction. We revised paragraph 3 of the Introduction (page 3, lines 3 to 18) to highlight the paradigm shift in T2DM management from a glucose-centric to one that focuses on cardiovascular risk reduction. Comment 2 - Eliminate the spaces between paragraphs on pages 3 and 4. Reduce the figure dimension. Response 2 – The spaces between paragraphs on pages 3 and 4 have been removed. The dimensions for all figures have been reduced. Please note that a new figure (figure 5a and b) has been added to reflect the differences in SGLT1 and SGLT2 expression. |
Reviewer 3 Report
Comments and Suggestions for Authors
In the present study, the authors presented an interesting review regarding the benefits of novel drugs for T2DM on cardio-vascular and renal disease. Please see my comments below.
· Page 3, line 2: To use “in the past decade” it's not quite right. First GLP-1 RA was approved in 2004/2005. I recommend a reformulation.
· Page 4, line 6: Authors said that the timeline for all searches was restricted to 10 years, but in the References section there are a lot of references published before this period. Please clarify this aspect. also, there are other recently papers could be included in the article (eg., https://doi.org/10.3390/ijms25020794, 10.1016/j.jchf.2023.08.026 etc.)
· Page 11: For a better understanding, the authors could present comparatively the places where SGLT1 and SGLT2 are expressed.
· The use of GLP-1 RAs in obese patients should be addressed
· Authors should present the risks of SGLT2 inhibitors and GLP-1 RAs (e.g.: https://doi.org/10.3389/fcvm.2022.1010693, doi: 10.20944/preprints202404.1954.v1, https://doi.org/10.1016/j.dx.2022.102427)
· Could you add some information regarding lixisenatide or exenatide?
· The drugs approved (e.g. tirzepatide) should be presented separately by the other studied in different phases of clinical trials. Please correct the Figure 6, too. Tirzepatide (GLP-1 RAs/GIP) is approved on the market.
· Please present the novelty of the study.
Author Response
|
Thank you very much for taking the time to review this manuscript. Please find the detailed responses below and the revisions/corrections highlighted in the re-submitted files. Comment 1: Page 3, line 2: To use “in the past decade” it's not quite right. First GLP-1 RA was approved in 2004/2005. I recommend a reformulation. Response 1: We agree that the use of “in the past decade” alone could lead to misinterpretation, as the first GLP-1 RA was approved two decades ago. We have amended page 3, lines 3 to 5, to highlight our intention to convey that it was in the past decade that we witnessed the completion of the landmark cardiovascular outcome trials for SGLT2 inhibitor empagliflozin and GLP-1 RA liraglutide. Comment 2 - Page 4, line 6: Authors said that the timeline for all searches was restricted to 10 years, but in the References section there are a lot of references published before this period. Please clarify this aspect. also, there are other recently papers could be included in the article (eg., https://doi.org/10.3390/ijms25020794, 10.1016/j.jchf.2023.08.026 etc.) Response 2 —Thank you for commenting on our literature search. The duration of the search for relevant clinical trials was limited to 10 years. However, we included several other references to explain key concepts behind, for example, some of the biomarkers studied in the clinical trials. We agree with the suggestion to include recent papers on heart failure. A new paragraph on page 7, lines 18-29, and lines 22-26 on page 9 were inserted to discuss SGLT2 inhibition in HFpEF. Comment 3 - Page 11: For a better understanding, the authors could present comparatively the places where SGLT1 and SGLT2 are expressed. Response 3—Thank you for the suggestion. We have changed the paragraph on dual SGLT1 and SGLT2 inhibitors on page 12 and added a new figure (Figure 5a and b) to illustrate better the differences in the location and function of SGLT1 and SGLT2 receptors. Comment 4 - The use of GLP-1 RAs in obese patients should be addressed. Response 4 – Thank you for the suggestion. A paragraph on the use of GLP-1 RAs in obese patients has been added (lines 27-35 on page 13). Comment 5 - Authors should present the risks of SGLT2 inhibitors and GLP-1 RAs. Response 5 —Thank you for the suggestion. A paragraph on the adverse effects of SGLT2 inhibitors and GLP-1 RAs has been added to sections 3.5 (page 11, lines 14-31) and 4.9 (page 18, lines 5-15). Comment 6 - Could you add some information regarding lixisenatide or exenatide? Response 6 – Thank you for the suggestion. A paragraph on lixisenatide and exenatide has been added on page 13, lines 15 – 26. Comment 7 - The drugs approved (e.g. tirzepatide) should be presented separately by the other studied in different phases of clinical trials. Please correct the Figure 6, too. Tirzepatide (GLP-1 RAs/GIP) is approved on the market. Response 7 – Thank you for the comment. We have added a new table in supplemental data to summarise completed and ongoing clinical trials involving tirzepatide. The figure has been corrected. Comment 8 - Please present the novelty of the study. Response 8 – Thank you for the comment. The novelty of this review lies in providing a timeline to help readers visualize the past, present and future of T2DM management. We have made changes to paragraphs 3 and 4 of the introduction. We restructured them to provide a background of the paradigm shift in T2DM management and forecast what could be expected in the coming decade by appraising the future landscape of novel antidiabetic agents. |